# Collaborative attitudes and trust among medical and dental professionals in Saudi Arabia

Khalid Aboalshamat[1,¤a‡]*, Emad Alzahrani[2,¤b‡], Anas Maqlan[2,¤b‡], Adel Almatrafi[2,¤b‡], Abdulaziz Alsulami[2,¤b‡]

1 Faculty of Dental Medicine, Preventative Dentistry Department, Dental Public Health Division, Umm Al-Qura University, Makkah, Saudi Arabia, 2 College of Dentistry, Umm Al-Qura University, Makkah, Saudi Arabia

¤a Current address: Faculty of Dental Medicine, Preventative Dentistry Department, Umm Al-Qura University, Makkah, Saudi Arabia
¤b Current address: Faculty of Dental Medicine, Umm Al-Qura University, Makkah, Saudi Arabia
‡ The following authors contributed equally to this work in the research idea, protocol development, data collection, data analysis, and manuscript writing: KA, EA, AM, AdA, and AbA
* Ktaboalshamat@uqu.edu.sa

**Data Availability Statement:** All relevant data are within the paper and its Supporting Information files.

## Abstract

### Background

Interprofessional collaboration involves coordinated efforts by individuals from different professions. Trust is defined as an individual's willingness to be vulnerable within a relationship, while expecting the other party to act in their best interests. This study aimed to assess the attitudes of collaboration and trust among dental and medical professionals in Saudi Arabia.

### Materials and methods

This cross-sectional study evaluated the attitudes of collaboration and trust among dental and medical professionals in Saudi Arabia using an online self-reported questionnaire distributed via social media. Data analysis was performed using SPSS.

### Results

The overall mean score for collaborative attitude was 6.2 (SD = 1.52), with 69.23% of participants exhibiting a positive attitude about collaboration. Participants rated family medicine (76.67%) and pediatric medicine (76.41%) as the medical specialties most relevant to dentistry, while obstetrics/gynecology was rated the lowest (32.31%). Dental professionals had a significantly higher mean score for collaborative attitude (m = 6.46, SD = 1.48) than did medical professionals (m = 5.93, SD = 1.51; p < 0.001). The score for dental professionals' trust in medical professionals (DTM) (m = 36.94, SD = 8.06) was significantly higher (p < 0.001) than the score for medical professionals' trust in dental professionals (MTD) (m = 33.81, SD = 8.20). The collaborative attitude scores among dental and medical professionals were not statistically significant when tested against the MTD score (p = 0.777) and DTM score (p = 0.419).

**Funding:** The author(s) received no specific funding for this work.

**Competing interests:** The authors have declared that no competing interests exist.

**Abbreviations:** MTD, Medical professionals' trust in dental professionals; DTM, Dental professionals' trust in medical professionals; SD, Standard deviation.

## Conclusions

There is a high level of collaborative attitude and trust between dental and medical professionals in Saudi Arabia. Dentists exhibit a higher degree of these attributes than medical professionals. These findings support the implementation of a collaborative medical–dental education framework in Saudi Arabia, where both specialties can train together during their undergraduate years.

## Introduction

Oral health is an important and essential factor in general health [1] and is frequently associated with underlying diseases [2]. Poor oral health can lead to various adverse effects on well-being, a decrease in the quality of life, and the initiation of a systemic inflammatory response [3–5]. Oral health is typically provided by dental health care workers; however, other health care professionals can also be involved in a team of interprofessional collaboration [6]. Interprofessional collaboration is defined as an effort to solve an issue or provide a service carried out by people from different professions [6]. From the perspective of primary care, it is crucial that physicians and dentists enhance collaboration and exchange information that might positively impact patient health [7], leading to improved health care services [8]. Although the interprofessional cooperation between general practitioners and dentists can sometimes be insufficient, it remains significant [9].

There are few articles that discuss the awareness of and attitudes about collaboration between dental and medical professionals [10–12]. Some of these studies found a positive attitude and awareness of the collaboration between dental and medical practices among students in Indonesia and Hong Kong [11,12]. One study found that senior students had a more positive attitude and awareness about the collaboration than junior students [11]. Also, female students were more aware of the collaboration than male students in Indonesia and India [10,11].

However, these studies showed controversial results regarding collaboration, with one of the studies reporting that dental students had a more positive attitude about collaboration than did medical students in Indonesia [11], while another study found the reverse in India [10]. The studies provided in-depth results; for example, 66% of the participants were aware of the link between dentistry and emergency medicine, but only 7% were aware of the link between dentistry and obstetrics/gynecology [11]. Also, 82% of the medical students believed that dental checkups should be included in health insurance packages [10], but their attitudes toward such collaborative practices diminished over their academic years [10]. In general, studies have indicated that collaboration between dental and medical professionals is viewed favorably in some countries, but these results cannot be generalized to other countries unless further studies are conducted.

From another perspective, trust is defined as an individual's willingness to expose a vulnerability in a relationship with the expectation that the other person will act in their best interests [13]. This involves communication between people or between people and a system [13]. Gaining trust is a crucial step in building a physician–patient or dentist–patient relationship [14,15]. Research has shown that people who trust their doctors are more likely to accept their treatment [16]. Another study demonstrated that a trusting relationship between dentists and patients through successful communication plays an essential role in managing patients' anxiety about dental treatment [17]. Building a trusting relationship in the first few minutes of a clinical encounter can enhance the treatment experience and reduce patient anxiety,

encouraging patient participation in treatment decisions [18–20]. Despite the importance of interprofessional collaboration between dental and medical professionals, studies assessing trust between these professionals are scarce.

## Aim

The aim of our study was to assess the collaborative attitudes and trust between dental and medical professionals in Saudi Arabia. It is noteworthy that in Saudi Arabia, medical doctors and dentists are trained separately as distinct professionals with different responsibilities.

## Materials and methods

This cross-sectional study aimed to estimate the collaboration and trust between dental and medical professionals. Sample size calculation was based on an alpha level of 0.05, a confidence interval of 95%, and an expected prevalence of 50%, and the minimum required sample size was estimated to be 385. The inclusion criteria for participation in this study included students, interns, or graduates from dental or medical colleges in Saudi Arabia or working doctors and dentists in Saudi Arabia. The exclusion criteria were retired participants from either specialty and those who did not sign the consent form. The study protocol was approved by the institutional review board of Umm Al-Qura University with the approval number HAPO-02-K-012-2023-11-1880. Electronic informed consent was obtained from all participants. A link to the informed consent form and study questionnaire was sent to potential participants, who were required to agree to the consent form before answering the questionnaire. Recruitment commenced on 25/12/2023 and concluded on 27/1/2024.

A convenience sample method was used for this study, with an electronic questionnaire for data collection. The study link was distributed through social media apps, such as Telegram, Instagram, WhatsApp, and X (previously known as Twitter). The invitation targeted dental and medical social media groups. In addition, the research team identified potential participants through a search of social media databases using specific keywords and then invited those candidates to the study.

The questionnaire comprised 29 questions divided into four sections. The first section contained seven demographic questions. The second section included eight questions aimed at assessing the collaboration between dental and medical professionals with yes/no questions. This section was adapted from a previous study [11], and the section's total score was the sum of yes answers, given one point each, to the eight questions. The total score was categorized as negative attitude (0 to 2 points), neutral attitude (3 to 5 points), or positive attitude (6 to 8 points).

The third section asked participants to assess which medical specialties (family medicine, pediatric medicine, emergency medicine, etc.) have links to dentistry. The fourth section included 11 questions about the level of trust between dental and medical professionals, adapted from a previous study with modifications [21]. Each question was answered on a Likert scale ranging from 1 (strongly disagree) to 5 (strongly agree). The modification was the adaptation of the questionnaire into two formats. The first measured medical professionals' trust in dental professionals (MTD), and the second measured dental professionals' trust in medical professionals (DTM), but they otherwise contained the same questions. The trust questionnaire's total score ranged from 11 to 55, as the highest level of trust. The total score was composed of the sum of all trust questionnaire responses, keeping in mind that some questions were scored in reverse (questions 2 and 7).

For data analysis, two software programs were utilized: Microsoft Excel software 2023 v.2309 (Microsoft Corp, Redmond, WA, USA) and SPSS v.29 (IBM, Armonk, NY, USA). The data were analyzed descriptively, generating means, standard deviation, count, and

percentages. Additionally, some statistical tests were used for analytical purposes, including t-test, ANOVA, Mann–Whitney U test, Kruskal–Wallis, and linear regression. A significance level of $p < 0.05$ was adopted.

## Results

A total of 390 participants answered the study questionnaire. Their demographic data are visualized in Table 1. The participants had a mean (m) score of 24.01 and standard deviation (SD) of 5.53.

The participant responses to the questions regarding attitudes about collaboration between dental and medical professionals are shown in Table 2. The mean collaboration score was 6.21 (SD = 1.52). When the total collaboration score was categorized, there were 4 respondents (1.02%) who had a negative attitude about collaboration, 116 (29.74%) who had a neutral attitude about collaboration, and 270 (69.23%) who had a positive attitude. According to a t-test, the collaborative attitude score among dental professionals (m = 6.46, SD = 1.48) was significantly higher ($p < 0.001$) than that found among medical professionals (m = 5.93, SD = 1.51). However, the simple linear regression, t-test, and ANOVA showed that the collaborative attitude score was not significantly associated with age ($p = 0.055$), gender ($p = 0.921$), qualifications ($p = 0.125$), nationality ($p = 0.323$), region ($p = 0.206$), or current place of work or study ($p = 0.282$). Participants had different ratings for the medical specialties that are related to dentistry, as shown in Table 2.

The participants responses to the trust questions (MTD) and (DTM) are shown in Tables 3 and 4. The t-test showed that the DTM score (m = 36.94, SD = 8.06) was significantly higher ($p < 0.001$) than the MTD score (m = 33.81, SD = 8.20).

The linear regression, t-test, Mann–Whitney U test, and Kruskal–Wallis test showed that the MTD and DTM scores were not statistically significant when tested against age, gender, qualification, nationality, or current place of work or study.

**Table 1. Participant demographic data.**

| Variable | | n | % |
|---|---|---|---|
| **Gender** | Male | 188 | 48.21% |
| | Female | 202 | 51.79% |
| **Specialty** | Medicine | 183 | 46.92% |
| | Dentistry | 207 | 53.08% |
| **Qualification** | Students in non-clinical years (years 1–3) | 122 | 31.28% |
| | Student in clinical years (years 4–6) | 141 | 36.15% |
| | Graduate (bachelor's degree) | 90 | 23.08% |
| | Specialist | 18 | 4.62% |
| | Consultant | 19 | 4.87% |
| **Region in Saudi Arabia** | Western | 189 | 48.46% |
| | Central | 86 | 22.05% |
| | Southern | 36 | 9.23% |
| | Eastern | 58 | 14.87% |
| | Northern | 21 | 5.38% |
| **Nationality** | Saudi | 370 | 94.87% |
| | Non-Saudi | 20 | 5.13% |
| **Current education or work sector** | Governmental | 318 | 81.54% |
| | Private | 46 | 11.79% |
| | Both | 26 | 6.67% |

**Table 2. Participant attitudes about collaboration between dental and medical professionals.**

| Variable | n (yes) | % (yes) |
|---|---|---|
| **Attitude statements** | | |
| Dentist is a profession similar to medical practitioners. | 278 | 71.28% |
| Oral health is an integral part of general health. | 367 | 94.10% |
| Dentists should be able to access and use the electronic health records systems. | 366 | 93.85% |
| Medical–dental collaboration enhances the quality of patient care. | 371 | 95.13% |
| A dentist is responsible for advising patients about systemic health. | 267 | 68.46% |
| A physician is responsible for advising patients about oral health. | 287 | 73.59% |
| Dental students should have a rotation in medicine. | 253 | 64.87% |
| Medical students should have a rotation in dentistry. | 235 | 60.26% |
| **Which of the following medical specialties have a collaboration (link) with dentistry?** | | |
| Family medicine | 299 | 76.67% |
| Pediatric medicine | 298 | 76.41% |
| Emergency medicine | 273 | 70.00% |
| Psychiatry and radiology | 256 | 65.64% |
| Orthopedics/traumatology | 232 | 59.49% |
| Clinical oncology | 222 | 56.92% |
| Otolaryngology | 221 | 56.67% |
| Cardiothoracic surgery | 215 | 55.13% |
| General surgery | 211 | 54.10% |
| Obstetrics/gynecology | 126 | 32.31% |

A simple linear regression revealed that the collaboration score of medical professionals was not statistically significant with the MTD score (p = 0.777). There was a similar relationship between the collaboration score of dental professionals and the DTM score (p = 0.419).

## Discussion

This study aimed to assess the collaborative attitudes and trust between dental and medical professionals in Saudi Arabia. A total of 69.23% of our respondents had a positive collaborative

**Table 3. Medical professionals' trust in dental professionals (MTD).**

| Item | m | SD |
|---|---|---|
| Dentists, in general, care about their patients' health just as much or more as their patients do. | 3.17 | 1.37 |
| Sometimes, dentists care more about what is convenient for them than about their patients' dental needs.* | 2.79 | 1.21 |
| Dentists are extremely thorough and careful. | 3.15 | 1.21 |
| You completely trust dentists' decisions about which dental treatments are best. | 3.23 | 1.37 |
| Dentists are totally honest in telling their patients about all of the different treatment options available for their condition. | 2.92 | 1.28 |
| Dentists think only about what is best for their patients. | 2.9 | 1.13 |
| Sometimes, dentists do not pay full attention to what patients are trying to tell them.* | 2.76 | 1.2 |
| Dentists always use their very best skill and effort on behalf of their patients. | 3.3 | 1.23 |
| You have no worries about putting your life in the hands of dentists. | 2.83 | 1.32 |
| A dentist would never mislead you about anything. | 2.84 | 1.17 |
| All in all, you trust dentists completely. | 3.02 | 1.23 |

*These items were scored in reverse when calculating the MTD score.

Table 4. Dental professionals' trust in medical professionals (DTM).

| Item | m | SD |
|---|---|---|
| Doctors, in general, care about their patients' health just as much or more as their patients do. | 3.48 | 1.25 |
| Sometimes, doctors care more about what is convenient for them than about their patients' medical needs.* | 2.84 | 1.2 |
| Doctors are extremely thorough and careful. | 3.49 | 1.16 |
| You completely trust doctors' decisions about which medical treatments are best. | 3.49 | 1.2 |
| Doctors are totally honest in telling their patients about all of the different treatment options available for their condition. | 3.39 | 1.18 |
| Doctors think only about what is best for their patients. | 3.38 | 1.19 |
| Sometimes, doctors do not pay full attention to what patients are trying to tell them.* | 2.81 | 1.1 |
| Doctors always use their very best skill and effort on behalf of their patients. | 3.51 | 1.09 |
| You have no worries about putting your life in the hands of doctors. | 3.38 | 1.24 |
| A doctor would never mislead you about anything. | 3.16 | 1.04 |
| All in all, you trust doctors completely. | 3.3 | 1.07 |

*These items were scored in reverse when calculating the DTM score.

attitude. The participants most often indicated that family medicine and pediatric medicine were medical specialties related to dentistry, while obstetrics/gynecology was the least frequently cited one. Dental professionals had significantly higher collaborative attitudes than those found among medical professionals. The trust scores of both medical (MTD) and dental professionals (DTM) in each other were higher than the midpoint of the scale (moderately high). However, the DTM score was significantly higher than the MTD score. The collaboration attitude scores among dental and medical professionals were not statistically significant with the MTD and DTM scores. Also, the collaborative attitude, DTM, and MTD were not statistically significant with age, gender, qualification, nationality, or current place of work or study.

The previous studies that investigated attitudes about collaboration between dental and medical personnel focused on students [10–12], while in our study, we expanded the population to include graduates, specialists, and consultants in dentistry and medicine. Our results and these previous studies agreed in showing that there are, in general, good attitudes about collaboration between dental and medical personnel. Nevertheless, there are some differences in terms of the demographic variables associated as the following. The previous studies reported that most students had a positive attitude about dental and medical collaboration [11,12], which is similar to our results. However, our results were aligned with a study from Indonesia [11] that found dental students had a more collaborative attitude than medical students. This is contradicted by another study in India [10], that indicated medical students had a more positive attitude. Our results showed no effect of gender on the attitude about collaboration, which contradicts previous studies, some of which found female students had a more positive attitude [10,11], while others found males had a more positive attitude [10]. Furthermore, previous studies found the level of students' education to be a significant variable in collaborative attitudes [11], but we did not find such a relationship in our study. This might be due to differences in cultures, educational systems, work environments, and the interaction between males and females in each country. The difference might also be due to our study including dental and medical workers in addition to students. However, we cannot have a conclusive explanation for this, and more studies are needed to validate this point to explain the differences between countries in finding such different results.

We found in our study that the medical specialties most frequently related to dentistry were family medicine (76.67%) and pediatric medicine (76.41%), but previous studies found (66%)

emergency medicine most frequently linked with dentistry [11], while ear, nose, and throat also had a close link to dentistry (53%) [12]. However, our study and previous studies agreed that the least frequently mentioned medical specialty related to dentistry was obstetrics/gynecology [11,12]. It seems that the link between obstetrics/gynecology and dentistry is not very clear to some health students and practitioners, but there is, in fact, a link. For example, it is well documented that there are gingival tissue changes during pregnancy [22,23]. Also, some health practitioners believed that a particular trimester is better for taking radiographs rather than the others [24], but the American Dental Association and the American Congress of Obstetricians and Gynecologists recommendation indicated that dental radiographs are safe if radiograph precautions are taken [25,26]. One notable finding was that not all dental professionals had a high level of knowledge regarding dental precautions for pregnant women [24], and there are some attempts to improve public knowledge about dental precautions during pregnancy [27]. This shows the importance of establishing collaboration and communication venues between dental and medical professionals, as they treat the same patients in different situations and conditions, and patients might get confused or overwhelmed if they receive different recommendations and lines of treatment.

In Saudi Arabia, the educations of dental and medical professionals are conducted separately, which could impact any collaboration between them. A literature review suggested having a program that connects the gap between dental and medical educational training, especially in senior years [28]. This would help students acquire a greater understanding of the relationship between oral health and general health, allowing them to determine and treat conditions that affect both, resulting in enhanced patient outcomes, better treatment of chronic diseases, and decreased health care charges [28].

Our results showed that medical professionals have a moderately high level of trust in dental professionals. Previous studies have assessed public trust in dental professionals and found acceptable levels of trust in dentists in different countries, including Australia, the United States, and Russia [14,29,30], as well as among an Arabic population [31]. Despite the similarities of our study to previously mentioned prior studies, it must be noted that the level of trust the public has for dental professionals might be different than the trust medical and dental professionals have for each other. This is due to the fact that public–dentist trust can be influenced by several factors in the clinical setting, such as experiencing embarrassment, pain, stimuli that provoke gagging, or socioeconomic status [32]. Conversely, the trust between dental and medical professionals involves more collaborative interactions on a professional level.

Also, our research results showed that dental professionals have moderately high levels of trust in medical professionals. Previous studies of the public and medical professionals' trust found acceptable levels of trust in doctors in different countries [33,34]. Research has suggested that trust in medical professionals might be affected by the doctor's behavior, observed comfort levels, and personal relationships with the other party [35].

In our results, dental professionals had significantly higher collaborative attitudes and trust in medical professionals than medical professionals had in dental professionals. This was not surprising, given that dental professionals need on some occasions medical clearance and approval of medically compromised patients and patients with systemic disease in order to proceed with treatment [36,37]. This might have made dental professionals more open to collaboration with and trust in medical professionals, while the reverse may not be common.

One of the interesting points in our results is that we did not find a relationship between trust and a collaborative attitude between dental and medical professionals, resulting in a rejection of our null hypothesis. This was not expected. In fact, it is more logical to collaborate with people you trust, as found in a previous study conducted in a different working environment [38]. The previous study also indicated that the relationship between trust and collaboration is

bidirectional [38]. The reason for our study finding no relationship between trust and collaboration is not clear. However, there are some suggested explanations, as follows. Treating a patient medically is a demanding task that is based on solid scientific evidence and medical guidelines. In other words, a dentist or medical professional has to collaborate for the sake of a patient's benefit, even if they lack trust or do not want to do so. Also, there are some responsibilities and tasks conducted by dental or medical professionals that cannot be done by the other party, which forces collaboration based on the privileges of each specialty. From another perspective, there is strong governance in the health system in Saudi Arabia due to the recent health care transformation in Saudi Arabia during the Vision 2030 mega projects [39]. This governance can induce collaboration between different health team members in favor of the patients' needs and quality of care. Nevertheless, more studies are needed to investigate and explain such phenomena more in-depth.

This might be one of the few studies that have assessed collaborative attitudes and trust between two different populations (dental and medical) and investigated each one's view toward the other. However, there are some challenges with validating our results. This includes the convenience sample used, which resulted in an unequal distribution of participants from each region of Saudi Arabia. This resulted in a low level of external validity for our results. Future studies can investigate the factors that may enhance collaboration and trust between dental and medical professionals. Also, more efforts should be directed to increasing awareness of the link between dental and medical professionals, especially in areas that were not frequently mentioned, such as cardiothoracic surgery, general surgery, and obstetrics/gynecology.

## Conclusion

Around two-thirds of the dental and medical professionals from Saudi Arabia in our study had a positive attitude about collaboration with each other, with a significantly higher positive attitude among dental professionals. Also, there were moderately high levels of trust between dental and medical professionals, but it was significantly higher among dental professionals. This finding can be used as a cornerstone to advocate for a collaborative medical–dental education environment in Saudi Arabia, where both specialties can train together during their undergraduate years.

## Supporting information

**S1 File. Study raw data.**
(XLS)

## Author Contributions

**Conceptualization:** Khalid Aboalshamat, Emad Alzahrani, Anas Maqlan, Adel Almatrafi, Abdulaziz Alsulami.

**Data curation:** Khalid Aboalshamat, Emad Alzahrani, Anas Maqlan, Adel Almatrafi, Abdulaziz Alsulami.

**Formal analysis:** Khalid Aboalshamat, Emad Alzahrani, Anas Maqlan, Adel Almatrafi, Abdulaziz Alsulami.

**Methodology:** Khalid Aboalshamat, Emad Alzahrani, Anas Maqlan, Adel Almatrafi, Abdulaziz Alsulami.

**Project administration:** Khalid Aboalshamat.

**Writing – original draft:** Khalid Aboalshamat, Emad Alzahrani, Anas Maqlan, Adel Almatrafi, Abdulaziz Alsulami.

**Writing – review & editing:** Khalid Aboalshamat, Emad Alzahrani, Anas Maqlan, Adel Almatrafi, Abdulaziz Alsulami.

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
