## [Decision Letter · Decision Letter 0]

27 May 2024

PONE-D-24-18470Collaborative attitudes and trust among medical and dental professionals in Saudi ArabiaPLOS ONE

Dear Dr. Aboalshamat,

Thank you for submitting your manuscript to PLOS ONE. After careful consideration, we feel that it has merit but does not fully meet PLOS ONE’s publication criteria as it currently stands. Therefore, we invite you to submit a revised version of the manuscript that addresses the points raised during the review process.

We look forward to receiving your revised manuscript.

Kind regards,

Mohmed Isaqali Karobari, BDS, MScD.Endo, MFDS. RCPS Glasg, Ph.D. Endo

Academic Editor

PLOS ONE

Journal Requirements:

**Additional Editor Comments:**

Dear Authors,

Kindly read all the comments carefully and carry on the revisions in the revised manuscript accordingly

Best regards and keep well

Reviewers' comments:

Reviewer's Responses to Questions

**Comments to the Author**

1. Is the manuscript technically sound, and do the data support the conclusions?

Reviewer #1: Yes

Reviewer #2: Yes

2. Has the statistical analysis been performed appropriately and rigorously? 

Reviewer #1: Yes

Reviewer #2: Yes

3. Have the authors made all data underlying the findings in their manuscript fully available?

Reviewer #1: Yes

Reviewer #2: Yes

4. Is the manuscript presented in an intelligible fashion and written in standard English?

Reviewer #1: Yes

Reviewer #2: Yes

5. Review Comments to the Author

Reviewer #1: Introduction

1. There is a lack of clarity and flow.

2. There is repetition of sentences, particularly regarding the importance of interprofessional collaboration and the differences in attitudes among students from different regions.

3. Statements like "some of these studies" and "one study found" are vague. It would be more impactful to specify which studies are being referred to

4. The introduction does not clearly define the term "interprofessional collaboration" until midway through the paragraph.

5. Insufficient Literature Review

6. There is a confusion as introduction shifts focus from general oral health to student attitudes without a clear rationale.

7. Kindly ensure that all references are correctly formatted and consistently presented. For example, “1” should be verified for proper citation style according to the journal's guidelines.

Aim

1. The clear connection is missing from the concept of trust to interprofessional collaboration.

2. There is some repetition in the explanation of trust and its importance in patient relationships.

Materials and Methods Section

1. The materials and methods section is very brief and lacks details regarding information on how the sample size was calculated, the sampling method, data collection procedures, and analysis techniques.

2. The methods would be made better by breaking it down into study design, setting, participants, sample size calculation, data collection, and data analysis.

Results

1. There is bias in Convenience Sampling as it does not have a representative sample of the population.

2. The decision to rely on social media could be wrong as apps may exclude professionals who are not active on these platforms leading to a selection bias.

3. The breaking of category of attitudes based on scores (negative, neutral, positive) is somewhat random and may oversimplify complex attitudes.

Discussion

1. The discussion section presents the results but does not dig deep into the implications or reasons behind these findings.

2. The previous studies are mentioned but does not adequately compare and contrast these with the current study’s findings.

3. There is no thorough integration of the current study’s findings with the broader body of literature.

4. The discussion briefly mentions non-significant results (e.g., no significant differences based on age, gender, etc.) but does not explore their implications.

5. The discussion does not address the practical implications of the findings or provide recommendations for future practice or research.

6. The study lacks explanation for why no relationship was found between education level and collaborative attitudes in this study.

7. There is a clear lack of practical implications of the findings.

Reviewer #2: 1. The use of convenience sampling may introduce bias, as it does not guarantee a representative sample of the entire population of medical and dental professionals in Saudi Arabia. This impacts the external validity and generalizability of the findings.

2. Relying on social media for distributing the questionnaire might exclude professionals who are not active on these platforms, leading to potential sampling bias.

3. The study relies on self-reported data, which can be influenced by social desirability bias. Participants might provide answers they perceive as favourable rather than their true opinions.

4. The sample was unevenly distributed across different regions of Saudi Arabia, with a significant proportion from the Western region. This regional bias may affect the generalizability of the findings across the entire country.

5. The questionnaire was adapted from previous studies; there is limited information on whether the adapted version was validated for the Saudi context. Cultural differences might affect how questions are interpreted and answered.

6. The cross-sectional design limits the ability to infer causality. Longitudinal studies would be better suited to assess changes in attitudes and trust over time and to establish causal relationships.

7. The study identifies specific specialties that participants perceive as related to dentistry. However, it does not deeply explore the nature or quality of these interactions, nor does it identify strategies to enhance collaboration in less frequently mentioned specialties like obstetrics/gynecology.

8. Likert scales are useful for quantifying attitudes, they may oversimplify complex opinions and lead to central tendency bias.

9. Conducting longitudinal studies can help track changes in attitudes and trust over time, providing a deeper understanding of how these relationships evolve.

10. Ensuring that questionnaires are culturally adapted and validated for the specific context of Saudi Arabia can improve the reliability and validity of the results.

11. Incorporating qualitative methods, such as interviews or focus groups, can provide richer insights into the reasons behind the attitudes and trust levels, as well as potential barriers to effective collaboration.

12. Investigating the impact of integrated educational programs on the collaborative attitudes and trust between dental and medical students could provide evidence for curriculum development.

13. The study provides valuable insights into the collaborative attitudes and trust between dental and medical professionals in Saudi Arabia, addressing its limitations through methodological improvements and further research can enhance the understanding and effectiveness of interprofessional collaboration in healthcare.

6. PLOS authors have the option to publish the peer review history of their article (what does this mean?). If published, this will include your full peer review and any attached files.

Reviewer #1: **Yes: **Dr Sandeep Bailwad

Reviewer #2: No

---

## [Author Response · Author response to Decision Letter 0]

21 Jul 2024

All comments were adjusted as found in the attached file.

---

## [Decision Letter · Decision Letter 1]

29 Jul 2024

Collaborative attitudes and trust among medical and dental professionals in Saudi Arabia

PONE-D-24-18470R1

Dear Dr. Aboalshamat,

We’re pleased to inform you that your manuscript has been judged scientifically suitable for publication and will be formally accepted for publication once it meets all outstanding technical requirements.

Kind regards,

Prof Dr Mohmed Isaqali Karobari, BDS, MScD. Endo, MFDS. RCPS Glasg, FDS. RCS Eng, Ph.D. Endo

Academic Editor

PLOS ONE

Additional Editor Comments (optional):

Dear Authors,

The authors have addressed all the comments and suggestions reviewers gave, and the manuscript has dramatically improved. I would like to congratulate the authors and wish them all the very best in their future endeavors.

Best regards and keep well

Reviewers' comments:

Reviewer's Responses to Questions

**Comments to the Author**

1. If the authors have adequately addressed your comments raised in a previous round of review and you feel that this manuscript is now acceptable for publication, you may indicate that here to bypass the “Comments to the Author” section, enter your conflict of interest statement in the “Confidential to Editor” section, and submit your "Accept" recommendation.

Reviewer #1: All comments have been addressed

Reviewer #2: All comments have been addressed

2. Is the manuscript technically sound, and do the data support the conclusions?

Reviewer #1: Yes

Reviewer #2: Yes

3. Has the statistical analysis been performed appropriately and rigorously? 

Reviewer #1: Yes

Reviewer #2: Yes

4. Have the authors made all data underlying the findings in their manuscript fully available?

Reviewer #1: Yes

Reviewer #2: Yes

5. Is the manuscript presented in an intelligible fashion and written in standard English?

Reviewer #1: Yes

Reviewer #2: Yes

6. Review Comments to the Author

Reviewer #1: Thank you Authors for addressing all the recommendations made and making the necessary corrections

Please keep up the good work

Reviewer #2: Dear Authors, Thank you for addressing all the recommendation as submitted for the manuscript. Please keep up the good work.

7. PLOS authors have the option to publish the peer review history of their article (what does this mean?). If published, this will include your full peer review and any attached files.

Reviewer #1: **Yes: **Dr Sandeep Bailwad

Reviewer #2: No

---

## [Editor Report · Acceptance letter]

20 Aug 2024

PONE-D-24-18470R1 

PLOS ONE

Dear Dr. Aboalshamat, 

I'm pleased to inform you that your manuscript has been deemed suitable for publication in PLOS ONE. Congratulations! Your manuscript is now being handed over to our production team.

Kind regards, 

on behalf of

Prof Dr. Mohmed Isaqali Karobari 

Academic Editor

PLOS ONE